# Read and Reap the Rewards:
# Learning to Play Atari with the Help of Instruction Manuals

**Yue Wu**[1*], **Yewen Fan**[1], **Paul Pu Liang**[1], **Amos Azaria**[2], **Yuanzhi Li**[1,3], **Tom Mitchell**[1]

[1]Carnegie Mellon University, [2]Ariel University, [3]Microsoft Research

*ywu5@andrew.cmu.edu

## Abstract

High sample complexity has long been a challenge for RL. On the other hand, humans learn to perform tasks not only from interaction or demonstrations, but also by reading unstructured text documents, e.g., instruction manuals. Instruction manuals and wiki pages are among the most abundant data that could inform agents of valuable features and policies or task-specific environmental dynamics and reward structures. Therefore, we hypothesize that the ability to utilize human-written instruction manuals to assist learning policies for specific tasks should lead to a more efficient and better-performing agent. We propose the Read and Reward framework. Read and Reward speeds up RL algorithms on Atari games by reading manuals released by the Atari game developers. Our framework consists of a QA Extraction module that extracts and summarizes relevant information from the manual and a Reasoning module that evaluates object-agent interactions based on information from the manual. An auxiliary reward is then provided to a standard A2C RL agent, when interaction is detected. Experimentally, various RL algorithms obtain significant improvement in performance and training speed when assisted by our design. Code at github.com/Holmeswww/RnR.

## 1 Introduction

Reinforcement Learning (RL) has achieved impressive performance in a variety of tasks, such as Atari (Mnih et al., 2015; Badia et al., 2020), Go (Silver et al., 2017), or autonomous driving (Fuchs et al., 2021; Kiran et al., 2022), and is hypothesized to be an important step toward artificial intelligence (Silver et al., 2021). However, RL still faces a great challenge when applied to complicated, real-world-like scenarios (Shridhar et al., 2020b; Szot et al., 2021) due to its low sample efficiency. The Skiing game in Atari, for example, requires the skier (agent) to ski down a snowy hill and hit the objective gates in the shortest time, while avoiding obstacles. Such an intuitive game with simple controls (left, right) still required 80 billion frames to solve with existing RL algorithms (Badia et al., 2020), roughly equivalent to 100 years of non-stop playing.

Observing the gap between RL and human performance, we identify an important cause: the lack of knowledge and understanding about the game. One of the most available sources of information, that is often used by humans to solve real-world problems, is unstructured text, e.g., books, instruction manuals, and wiki pages. In addition, unstructured language has been experimentally verified as a powerful and flexible source of knowledge for complex tasks (Tessler et al., 2021). Therefore, we hypothesize that RL agents could benefit from reading and understanding publicly available instruction manuals and wiki pages.

37th Conference on Neural Information Processing Systems (NeurIPS 2023).

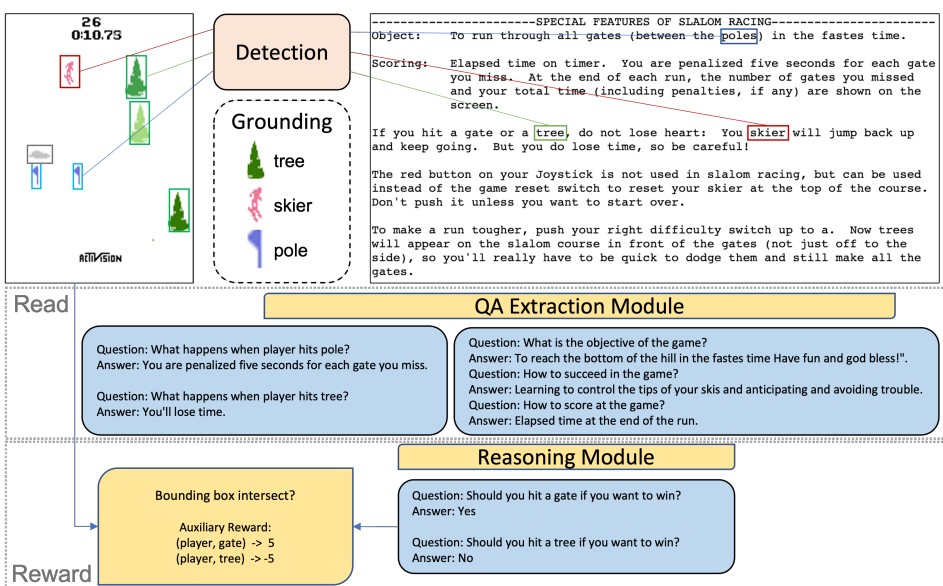

Figure 1: An overview of our Read and Reward framework. Our system receives the current frame in the environment, and the instruction manual as input. After object detection and grounding, the QA Extraction Module extracts and summarizes relevant information from the manual, and the Reasoning Module assigns auxiliary rewards to detected in-game events by reasoning with outputs from the QA Extraction Module. The "Yes/No" answers are then mapped to $+5/-5$ auxiliary rewards.

In this work, we demonstrate the possibility for RL agents to benefit from human-written instruction manuals. We download per-game instruction manuals released by the original game developers[1] or Wikipedia and experiment with 4 Atari games where the object location ground-truths are available.

Two challenges arise with the use of human-written instruction manuals: 1) *Length*: Manuals contain a lot of redundant information and are too long for current language models. However, only a few paragraphs out of the 2-3 page manuals contain relevant information for any specific aspect of concern, i.e., the game's objective or a specific interaction. 2) *Reasoning*: References may be implicit and attribution may require reasoning across paragraphs. For example, in Skiing, the agent needs to understand that hitting an obstacle is bad using both "you lose time if the skier hits an obstacle" and "the objective is to finish in shortest time", which are from different parts of the manual. Furthermore, the grounding of in-game objects to references in the text remains an open problem and manuals may contain significant syntactic *variation*. For example, the phrase "don't hit an obstacle" is also referred to as "don't get wrapped around a tree" or "avoid obstacles". Finally, the grounding of in-game features to references in the text remains an open problem (Luketina et al., 2019).

Our proposed Read and Reward framework (Figure 1) addresses the challenges with a two-step process. First, a zero-shot extractive QA module is used to extract and summarize relevant information from the manual, and thus tackles the challenge of *Length*. Second, a zero-shot reasoning module, powered by a large language model, *reasons* about contexts provided by the extractive QA module and assigns auxiliary rewards to in-game interactions. After object detection and localization, our system registers an interaction based on the distance between objects. The auxiliary reward from the interaction can be consumed by any RL algorithm.

Even though we only consider "hit" interactions (detected by tracking distance between objects) in our experiments, Read and Reward still achieves 60% performance improvement on a baseline using 1000x fewer frames than the SOTA for the game of Skiing. In addition, since Read and Reward does not require explicit training on synthetic text (Hermann et al., 2017; Chaplot et al., 2018; Janner et al., 2018; Narasimhan et al., 2018; Zhong et al., 2019, 2021; Wang & Narasimhan, 2021), we observe consistent performance on manuals from two *different* sources: Wikipedia and Official manuals.

To our knowledge, our work is the first to demonstrate the capability to improve RL performance in an end-to-end setting using real-world manuals designed for human readers.

---

[1] atariage.com

## 2 Background

### 2.1 Reducing sample complexity in RL

Prior works have attempted to improve sample efficiency with new exploration techniques (Schulman et al., 2017; Haarnoja et al., 2018; Badia et al., 2020), self-supervised objectives (Pathak et al., 2017; Schrittwieser et al., 2020), and static demonstration data (Kumar et al., 2019, 2020). Fan & Xiao (2022) cast RL training into a training data distribution optimization problem, and propose a policy mathematically controlled to traverse high-value and non-trivial states. However, such a solution requires manual reward shaping. Currently, the SOTA RL agent (Badia et al., 2020) still spends billions of frames to learn the Skiing game that can be mastered by humans within minutes.

### 2.2 Grounding objects for control problems

Most prior works study grounding – associating perceived objects with appropriate text – in the setting of step-by-step instruction following for object manipulation tasks (Wang et al., 2016; Bahdanau et al., 2018) or indoor navigation tasks (Chaplot et al., 2018; Janner et al., 2018; Chen et al., 2019; Shridhar et al., 2020a). The common approach has been to condition the agent's policy on the embedding of both the instruction and observation (Mei et al., 2016; Hermann et al., 2017; Janner et al., 2018; Misra et al., 2017; Chen et al., 2019; Pashevich et al., 2021). Recently, modular solutions have been shown to be promising (Min et al., 2021). However, the majority of these works use synthetic language. These often take the form of templates, e.g., "what colour is <object> in <room>" (Chevalier-Boisvert et al., 2018). On the other hand, recent attempts on large-scale image-text backbones, e.g., CLIP (Radford et al., 2021) have shown robustness at different styles of visual inputs, even on sketches. Therefore, we pick CLIP as our model for zero-shot grounding.

### 2.3 Reinforcement learning informed by natural language

In the language-assisted setting, step-by-step instructions have been used to generate auxiliary rewards, when environment rewards are sparse. Goyal et al. (2019); Wang et al. (2019) use auxiliary reward-learning modules trained offline to predict whether trajectory segments correspond to natural language annotations of expert trajectories.

In a more general setting, text may contain both information about optimal policy and environment dynamics. Branavan et al. (2012) improve Monte-Carlo tree search planning by accessing a natural language manual. They apply their approach to the first few moves of the game Civilization II, a turn-based strategy game. However, many of the features they use are handcrafted to fit the game of Civilization II, which limits the generalization of their approach.

Narasimhan et al. (2018); Wang & Narasimhan (2021) investigate planning in a 2D game environment with a fixed number of entities that are annotated by natural language (e.g. the 'spider' and 'scorpion' entities might be annotated with the descriptions "randomly moving enemy" and "an enemy who chases you", respectively). The agent learns to generalize to new environment settings by learning the correlation between the annotations and the environmental goals and mechanics. The proposed agent achieves better performance than the baselines, which do not use natural language. However the design of the 2D environment uses 6-line descriptions created from templates. Such design oversimplifies the observations into a 2D integer matrix, and removes the need for visual understanding and generalization to new objects and new formats of instruction manual, e.g., human written manuals.

### 2.4 RL Models that Read Natural Language Instructions

Zhong et al. (2019, 2021) make use of special architectures with Bidirectional Feature-wise Linear Modulation Layers that support multi-hop reasoning on $6 \times 6$ grid worlds with template-generated instruction manuals. However, the model requires 200 million training samples from templates identical to the test environments. Such a training requirement results in performance loss even on $10 \times 10$ grid worlds with identical mechanics, thus limiting the generalization of the model.

Wang & Narasimhan (2021) mixes embedding of entities with multi-modal attention to get entity representations, which are then fed to a CNN actor. The attention model and CNN actor are trained jointly using RL. However, the whole framework is designed primarily for a grid world with a template generated instruction manual consisting of one line per entity. In our experiments, we find

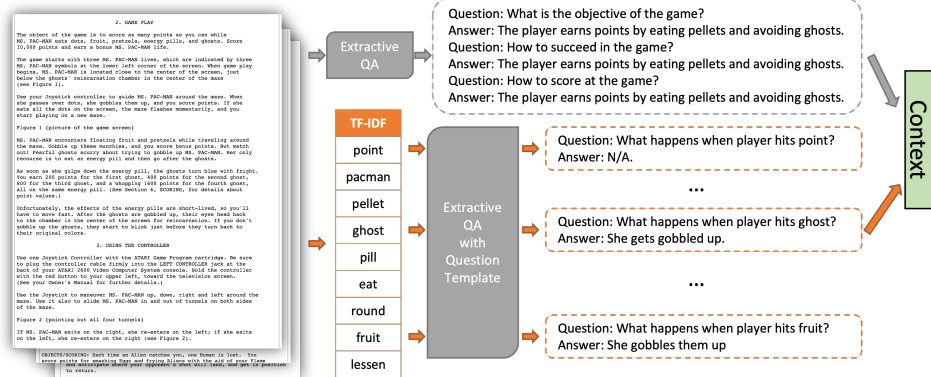

Figure 2: Illustration of the QA Extraction Module on the game PacMan. We obtain generic information about the game by running extractive QA on 4 generic questions (3 shown since one question did not have an answer). We then obtain object-specific information using a question template. We concatenate generic and object-specific information to obtain the <context> string.

that extracting entity embeddings from human-written Atari instruction manuals following Wang & Narasimhan (2021) does not improve the performance of our vision-based agent due to the complexity and diversity of the language.

# 3 Read and Reward

We identify two key challenges to understanding and using the instruction manual with an RL agent.

The first challenge is **Length**. Due to the sheer length of the raw text ($2 \sim 3$ pages), current pre-trained language models cannot handle the raw manual due to input length constraints. Additionally, most works in the area of long-document reading and comprehension (Beltagy et al., 2020; Ainslie et al., 2020; Zemlyanskiy et al., 2021) require a fine-tuning signal from the task, which is impractical for RL problems since the RL loss already suffers from high variance. Furthermore, an instruction manual may contain a lot of task-irrelevant information. For example, the official manual for MsPacman begins with: "MS. PAC-MAN and characters are trademark of Bally Midway Mfg. Co. sublicensed to Atari, Inc. by Namco-America, Inc."

It is therefore intuitive in our first step to have a QA Extraction Module that produces a summary directed toward game features and objectives, to remove distractions and simplify the problem of reasoning.

The second challenge is **Reasoning**. Information in the manual may be implicit. For example, the Skiing manual states that the goal is to arrive at the end in the fastest time, and that the player will lose time when they hit a tree, but it never states directly that hitting a tree reduces the final reward. Most prior works (Eisenstein et al., 2009; Narasimhan et al., 2018; Wang & Narasimhan, 2021; Zhong et al., 2019, 2021) either lack the capability of multi-hop reasoning or require extensive training to form reasoning for specific manual formats, limiting the generalization to real-world manuals with a lot of variations like the Atari manuals.

On the other hand, large language models have achieved success without training in a lot of fields including reading comprehension, reasoning, and planning (Brown et al., 2020; Kojima et al., 2022; Ahn et al., 2022). Therefore, in the Reasoning Module, we compose natural language queries about specific object interactions in the games, to borrow the in-context reasoning power of large language models.

The rest of the section describes how the two modules within our Read and Reward framework (a QA extraction module and a reasoning module) are implemented.

**QA Extraction Module (Read)**    In order to produce a summary of the manual directed towards game features and objectives, we follow the extractive QA framework first proposed by Devlin et al. (2018), which extracts a text sub-sequence as the answer to a question about a passage of text. The extractive QA framework takes raw text sequence $S_{\text{manual}} = \{w^0, ..., w^L\}$ and a question string $S_{\text{question}}$ as input. Then for each token (word) $w^i$ in $S_{\text{manual}}$, the model predicts the probability that the current token is the start token $p^i_{\text{start}}$ and end token $p^i_{\text{end}}$ with a linear layer on top of word piece embeddings. Finally, the output $S_{\text{answer}} = \{w^{\text{start}}, ..., w^{\text{end}}\}$ is selected as a sub-sequence of $S_{\text{manual}}$ that maximizes the overall probability of the start and end: $w^{\text{start}}, w^{\text{end}} = \arg\max_{w^{\text{start}}, w^{\text{end}}} p^{\text{start}} p^{\text{end}}$.

Implementation-wise, we use a RoBERTa-large model (Liu et al., 2019) for Extractive QA fine-tuned on the SQUAD dataset (Rajpurkar et al., 2016). Model weights are available from the AllenNLP API[2]. To handle instruction manual inputs longer than the maximum length accepted by RoBERTa, we split the manual into chunks according to the max-token size of the model, and concatenate extractive QA outputs on each chunk.

As shown in Figure 2, we compose a set of generic prompts about the objective of the game, for example, "What is the objective of the game?". In addition, to capture information on agent-object interactions, we first identify the top 10 important objects using TFIDF. Then for each object, we obtain an answer to the generated query:"What happens when the player hit a <object>?".

Finally, we concatenate all non-empty question-answer pairs per interaction into a <context> string. Note that the format of context strings closely resembles that of a QA dialogue.

**Zero-shot reasoning with pre-trained QA model (Reward)** The summary <context> string from the QA Extraction module could directly be used as prompts for reasoning through large language models (Brown et al., 2020; Kojima et al., 2022; Ahn et al., 2022).

Motivated by the dialogue structure of the summary, and the limit in computational resources, we choose Macaw (Tafjord & Clark, 2021), a general-purpose zero-shot QA model with performance comparable to GPT-3 (Brown et al., 2020). Compared to the RoBERTa extractive QA model for the QA Extraction module, Macaw is significantly larger and is better suited for reasoning. Although we find that GPT-3 generally provides more flexibility and a better explanation for its answers for our task, Macaw is faster and open-source.

As shown in Figure 3, for each interaction we compose the query prompt as "<context> Question: Should you hit a <object of interaction> if you want to win? Answer: ", and calculate the LLM score on choices {Yes, No}. The Macaw

Figure 3: Illustration of the Reasoning Module. The <context> (from Figure 2) related to the object **ghost** from the QA Extraction module is concatenated with a template-generated question to form the zero-shot in-context reasoning prompt for a Large Language Model. The Yes/No answer from the LLM is then turned into an auxiliary reward for the agent.

model directly supports scoring the options in a multiple-choice setting. We note that for other generative LLMs, we can use the probability of "Yes" vs. "No" as scores, following Ahn et al. (2022).

Finally, during game-play, a rule-based algorithm detects 'hit' events by checking the distance between bounding boxes. For detected interactions, an auxiliary reward is provided to the RL algorithm according to {Yes, No} rating from the reasoning module (see Section 4.4 for details).

# 4 Experiments

## 4.1 Atari Environment and Baselines

The Openai gym Atari environment contains diverse Atari games designed to pose a challenge for human players. The observation consists of a single game screen (frame): a 2D array of 7-bit pixels, 160 pixels wide by 210 pixels high. The action space consists of the 18 discrete actions defined by the joystick controller. Instruction manuals released by the original game developers have been scanned and parsed into publicly available HTML format[3]. For completeness, we also attach an explanation of the objectives for each game in Appendix A.3.

---

[2]AllenNLP Transformer-QA

[3]atariage.com

## 4.2 Delayed Reward Schedule

Most current Atari environments include a dense reward structure, i.e., the reward is obtained quite often from the environment. Indeed, most RL algorithms for the Atari environments perform well due to their dense reward structure. However, in many real-world scenarios (Kolve et al., 2017; Wang et al., 2022a) or open-world environments (Fan et al., 2022), it is expensive to provide a dense reward. The reward is usually limited and "delayed" to a single positive or negative reward obtained at the very end of an episode (e.g., once the robot succeeds or fails to pour the coffee).

Notably, Atari games with the above "delayed" reward structure, such as Skiing, pose great challenges for current RL algorithms, and are among some of the hardest games to solve in Atari (Badia et al., 2020). Therefore, to better align with real-world environments, we use a *delayed reward* version of the tennis, breakout, and Pac-Man games by providing the reward (the "final game score") only at the end of the game. This delayed reward structure does not change the optimization problem, but is more realistic and imposes additional challenges to RL algorithms.

## 4.3 Grounding Objects in Atari

We find grounding (detecting and relating visual objects to keywords from TFIDF) a significant challenge to applying our framework. In some extensively studied planning/control problems, such as indoor navigation, pre-trained models such as masked RCNN reasonably solve the task of grounding (Shridhar et al., 2020a). However, the objects in the Atari environments are too different from the training distribution for visual backbones like mask RCNN. Therefore, the problem becomes very challenging for the domain of Atari. Since it is not the main focus of our proposed work, we attempt to provide an unsupervised end-to-end solution only for the game of Skiing.

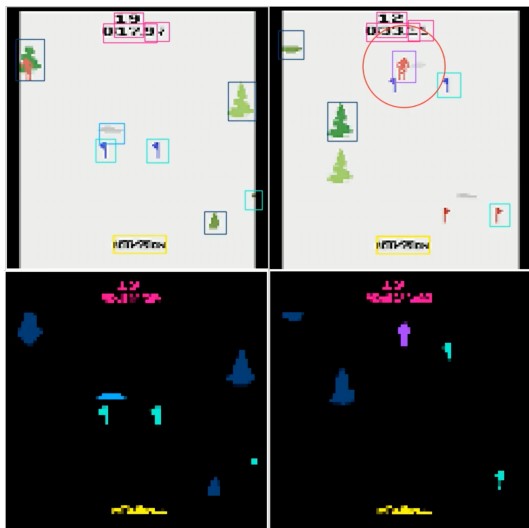

Figure 4: Examples of SPACE (Lin et al., 2020) and CLIP (Radford et al., 2021) in the full end-to-end pipeline (Section 4.3). The top row shows bounding boxes for objects and the bottom row shows corresponding object masks as detected by SPACE. Most of the bounding boxes generated are correct. **Left**: SPACE confuses bounding boxes of agent and tree into one and the box gets classified as "tree" (blue), and the auxiliary penalty is not properly triggered. **Right**: The flag next to the agent (in red circle) is not detected, and therefore the auxiliary reward is not provided.

**Full end-to-end pipeline on Skiing with A2C** To demonstrate the full potential of our work, we offer a proof-of-concept agent directly operating on raw visual game observations and the instruction manual for the game of Skiing. For unsupervised object localization, we use SPACE, a method that uses spatial attention and decomposition to detect visual objects (Lin et al., 2020). SPACE is trained on 50000 random observations from the environment, and generates object bounding boxes. These bounding boxes are fed to CLIP, a zero-shot model that can pair images with natural language descriptions (Radford et al., 2021), to classify the bounding box into categories defined by key-words from TF-IDF. The SPACE/CLIP models produce mostly reasonable results; however, in practice we find that CLIP lacks reliability over time, and SPACE cannot distinguish objects that cover each other. Therefore, to improve classification reliability, we use a Multiple Object Tracker (Bewley et al., 2016) in conjunction with CLIP and set the class of the bounding box to be the most dominant class over time.

**Ground-truth object localization/grounding experiments** For other games (Tennis, Ms. Pac-Man, and Breakout), we follow Anand et al. (2019) and directly calculate the coordinates of all game objects from the simulator RAM state. Specifically, we first manually label around 15 frames with object name and X,Y coordinates and then train a linear regression model mapping from simulator RAM state to labeled object coordinates. Note that this labeling strategy fails for games in which multiple objects of the same kind appears, i.e., multiple trees appear in one frame of Skiing.

|          | A2C+R&R    | A2C  | PPO+R&R    | PPO  | R2D1+R&R     | R2D1 | Agent57+R&R  | Agent57 |
|----------|------------|------|------------|------|--------------|------|--------------|---------|
| Tennis   | -5 (-5)    | -23  | -6 (-6)    | -23  | -1 (-1)      | -23  | 0 (0)        | -23     |
| Pacman   | 580 (580)  | 452  | 253 (253)  | 204  | 3115 (3115)  | 2116 | 1000 (1000)  | 708     |
| Breakout | 14 (14)    | 2    | 9 (9)      | 2    | 10 (10)      | 2    | 38 (38)      | 2       |

Table 1: Table of the game score of different RL algorithms trained under *delayed reward* schedule, RL with Read and Reward consistently outperforms their counterparts that are not using the manual. In addition, Read and Reward performance remains consistent across official Atari manual and Wikipedia in brackets "(.)".

|          | A2C+R&R        | A2C             | PPO+R&R         | PPO             | R2D1+R&R          | R2D1               |
|----------|----------------|-----------------|-----------------|-----------------|-------------------|--------------------|
| Tennis   | $-5.2 \pm 2.2$ | $-23 \pm 1.3$   | $-8.1 \pm 2.3$  | $-23 \pm 1.0$   | $-2.2 \pm 3.1$    | $-23.0 \pm 1.3$    |
| Pacman   | $455.2 \pm 63.2$ | $387.1 \pm 66.2$ | $-284.2 \pm 67.3$ | $200.1 \pm 65.4$ | $3001.3 \pm 203.2$ | $1999.2 \pm 109.2$ |
| Breakout | $-4.5 \pm 3.0$ | $2.1 \pm 0.3$   | $-10.2 \pm 0.1$ | $1.9 \pm 0.5$   | $10 \pm 3.0$      | $2.1 \pm 0.2$      |

Table 2: Table of the game score of different RL algorithms trained under *delayed reward* schedule on the official Atari manual, with statistics computed over 3 random seeds. RL with Read and Reward consistently outperforms their counterparts that are not using the manual.

## 4.4 Simplifying Interaction Detection to Distance Tracking

We make the assumption that all object interactions in Atari can be characterized by the distance between objects, and we ignore all interactions that do not contain the agent. We note that since the Atari environment is in 2D, it suffices to consider only interactions where objects get close to the agent so that we can query the manual for "hit" interaction.

We, therefore, monitor the environment frames and register interaction whenever an object's bounding boxes intersect with the agent. For each detected interaction, we assign rewards from $\{-r_n, r_p\}$ accordingly as detailed in Section 3. Experimentally, we find that it suffices to set $r_n = r_p = 5$.

## 4.5 RL Baselines

Since Read and Reward only provides an auxiliary reward, it is compatible with most RL frameworks such as A2C, PPO, SAC, TD3, and Agent 57 (Mnih et al., 2016; Schulman et al., 2017; Haarnoja et al., 2018; Fujimoto et al., 2018; Badia et al., 2020). We use popular open-source implementations A2C,PPO,R2D1 (Stooke & Abbeel, 2019), and Agent57 (Hu, 2022).

## 4.6 Full
## Pipeline Results on Skiing with A2C

We compare the performance of our Read and Reward agent using the full pipeline to baselines in Figure 5, our agent is best on the efficiency axis and close to best on the performance axis. A2C equipped with Read and Reward outperforms A2C baseline and more complex methods like MuZero (Schrittwieser et al., 2020). We note a 50% performance gap between our agent and the SOTA Agent 57, but our solution does not require policy mixing and requires more than 10 times fewer frames to converge to performance 60% better than random.

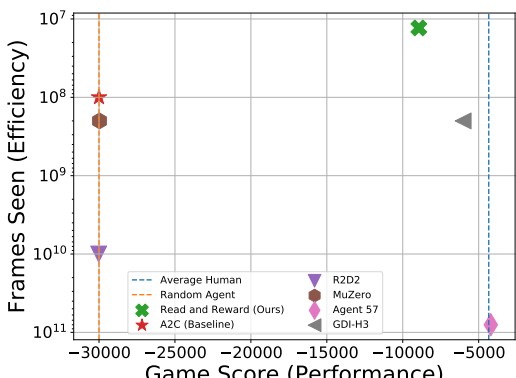

Figure 5: Performance (in Game Score) vs Efficiency (in Frames Seen) of fully end-to-end Read and Reward (green) compared to an A2C baseline, and MuZero (Schrittwieser et al., 2020) on Skiing. Benefiting from the auxiliary rewards from the instruction manual, Read and Reward out-performs the A2C Baseline and achieves 60% improvement on random performance while using much fewer training frames compared to the SOTA mixed-policy Agent 57 (Badia et al., 2020) and GDI-H3 (Fan & Xiao, 2022).

When experimenting with a full end-to-end pipeline as described in Section 4.3, we notice instability with the detection and grounding models. As shown in Figure 4, missing bounding boxes cause lots of mislabeling when objects are close to each other. Such errors may lead to the auxiliary reward being incorrectly assigned.

| Game | Wikipedia | Official Manual |
|---|---|---|
| | **What is the objective of the game?** | |
| | The player earns points by eating pellets and avoiding ghosts. | To score as many points as you can practice clearing the maze of dots before trying to gobble up the ghosts. |
| | **How to succeed in the game?** | |
| Pacman | The player earns points by eating pellets and avoiding ghosts. | Score as many points as you can. |
| | **How to score at the game?** | |
| | The player earns points by eating pellets and avoiding ghosts. | N/A |
| | **Who are your enemies?** | |
| | N/A | Ghosts. stay close to an energy pill before eating it, and tease the ghosts. |

Table 3: Table showing the outputs of the QA Extraction module on Wikipedia instructions vs the official Atari manual. The Wikipedia manual is significantly shorter, and contains less information, causing the extractive QA model to use the same answer for all questions. Full table for all 4 games is shown in Table 5 in the Appendix.

Although Skiing is arguably one of the easiest games in Atari for object detection and grounding, with easily distinguishable objects and white backgrounds, we still observe that current unsupervised object detection and grounding techniques lack reliability.

### 4.7 Results on Games with Delayed Reward Structure

Since the problem of object detection and grounding is not the main focus of this paper, we obtain ground-truth labels for object locations and classes as mentioned in Section 4.3. In addition, for a setting more generalizable to other environments (Kolve et al., 2017; Fan et al., 2022), we implement *delayed reward* schedule (Section 4.2) for Tennis, MsPacman, and Breakout. Various RL frameworks benefit significantly from Read and Reward, as shown in Table 1, 2.

We plot the Game Score and the auxiliary rewards of A2C with Read and Reward v.s. the A2C baseline in Figure 6. The A2C baseline fails to learn under the sparse rewards, while the performance of the Read and Reward agent continues to increase with more frames seen. In addition, we observe that the auxiliary reward from the Read and Reward framework has a strong positive correlation with the game score (game reward) of the agent, suggesting that the A2C agent benefits from optimizing the auxiliary rewards at training time.

### 4.8 Results without Delayed Reward

Without delayed reward, Read and Reward does not offer performance improvement in terms of game score, but still improves training speed. As shown in Table 4, Agent57 with Read and Reward requires up to 112X less training steps to reach the same performance as the Agent57 baseline without the instruction manuals.

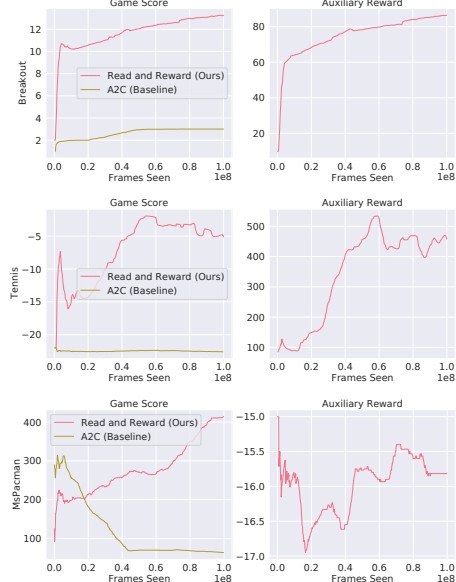

Figure 6: Comparison of the training curves of Read and Reward + A2C v.s. A2C Baseline in terms of game score alongside Auxiliary Reward v.s. Frames Seen under our delayed reward setting for 3 different Atari games. Read and Reward (red) consistently outperforms the A2C baseline (brown) in all games. In addition, the auxiliary reward from Read and Reward demonstrates a strong positive correlation with the game score, suggesting that the model benefits from optimizing the auxiliary rewards at training time.

### 4.9    R&R Behavior on Manuals from Different Source

To illustrate the generalization of our proposed framework to a different source of information, we download the "Gameplay" section from Wikipedia[4] for each game and feed the section as manual to our model. In Table 3, we show a comparison of the answers on the set of 4 generic questions on the Pacman by our QA Extraction Module. Note that the QA Extraction module successfully extracts all important information about avoiding ghosts and eating pellets to get points. In addition, since the official manual contains more information than the Gameplay section of Wikipedia (2 pages v.s. less than 4 lines), we were able to extract much more information from the official manual.

Due to the high similarity in the output from the QA Extraction Module, the Reasoning module demonstrates good agreement for both the Wiki manual and the official manual. Therefore, we observe consistent agent performance between the official manual and the Wikipedia manual in terms of game score (Table 1).

|  | Tennis | Pacman | Breakout |
|---|---|---|---|
| steps to reach baseline@1e6 | 6e5 | 5.1e5 | 8911 |
| speed-up ratio | 1.67 | 2 | 112 |

Table 4:  Table showing the number of training steps required by Agent57 with Read and Reward (Official) to reach the same performance as the Agent57 baseline at 1e6 training steps and the speedup ratio, under delayed-reward.

## 5    Limitations and Future Works

One potential challenge and limitations is the requirement of object localization and detection as mentioned in Section 4.3. However, since Read and Reward does not need to detect all objects to assign reward, i.e., energy pills are not grounded in the game Pacman but Read and Reward is still achieves improvement. In addition, reliable backbone object detection models (He et al., 2017) have already shown success in tasks such as indoor navigation (Shridhar et al., 2020b) and autonomous driving. With recent progress on visual-language models (Bubeck et al., 2023; Li et al., 2022; Driess et al., 2023; Liu et al., 2023; Zou et al., 2023), we believe that our proposed framework can benefit from reliable and generalizable grounding models in the foreseeable future.

Another simplification we made in Section 4.4 is to consider only interactions where objects get sufficiently **close** to each other. While such simplification appears to be sufficient in Atari, where observations are 2D, one can imagine this to fail in a 3D environment like Minecraft (Fan et al., 2022) or an environment where multiple types of interaction could happen between same objects (Kolve et al., 2017). Future works would benefit by grounding and tracking more types of events and interactions following recent works on video scene understanding Wang et al. (2022b) or game progress tracking Wu et al. (2023).

Finally, the performance of Read and Reward may be limited by information within the natural language data it reads.

## 6    Conclusions

In this work, we propose Read and Reward, a method that assists and speeds up RL algorithms on the Atari challenges by reading downloaded game manuals released by the Atari game developers. Our method consists of a QA extraction module that extracts and summarizes relevant information from the manual and a reasoning module that assigns auxiliary rewards to detected in-game events by reasoning with information from the manual. The auxiliary reward is then provided to standard RL agents. To our knowledge, this work is the first successful attempt for using instruction manuals in a fully automated and generalizable framework for solving the widely used Atari RL benchmark (Badia et al., 2020).

Our results suggest that even small open-source QA language models could extract meaningful gameplay information from human-written documentations for RL agents. With recent break-trough in zero-shot language understanding and reasoning with LLMs (Bubeck et al., 2023), we hope that our work could incentivize future works on integrating human prior knowledge into RL training.

---

[4]wikipedia.org

## Broader Impacts

Our research on using instruction manuals for RL could lead to both positive and negative impacts. The benefits include more efficient RL agents that could solve more real-world problems. The risks may involve more game cheating and exploitation, over reliance on the instruction manuals, and potential job loss.

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

# A Additional Details

## A.1 Scale of Auxiliary Rewards

In practice, the R&R framework is quite robust to the scale of auxiliary reward since reward clipping has been implemented by most RL algorithms (Hu, 2022). Any reward in the range of (2,50) should result in the same behavior.

## A.2 Atari Baselines

**A2C/PPO.** A2C (Mnih et al., 2016) and PPO (Schulman et al., 2017) are among some of the earliest Actor-Critic algorithms that brought success for deep RL on Atari games. A2C learns a policy $\pi$ and a value function $V$ to reduce the variance of the REINFORCE (Sutton et al., 1999) algorithm. The algorithm optimizes the advantage function $A(s,a) = Q(s,a) - V(s)$ instead of the action value function $Q(s,a)$ to further stabilize training.

**MuZero.** MuZero algorithm (Schrittwieser et al., 2020) combines a tree-based search with a learned model of the environment. The model of MuZero iteratively predicts the reward, the action-selection policy, and the value function. When applied to the Atari benchmarks, MuZero achieves super-human performance in a lot of games.

**R2D2/R2D1.** R2D2 (Kapturowski et al., 2018) proposes an improved training strategy for RNN-based RL agents with distributed prioritized experience replay. The algorithm is the first to exceed human-level performance in 52 of the 57 Atari games. R2D1 is a single-machine variant implemented by (Stooke & Abbeel, 2019).

**Agent57.** Badia et al. (2020) train a neural network that parameterizes a family of policies with different exploration, and proposes a meta-controller to choose which policy to prioritize during training. Agent57 exceeds human-level performance on all 57 Atari games.

## A.3 Explanation of Game Objectives

**Skiing.** The player controls the direction and speed of a skier and must avoid obstacles, such as trees and moguls. The goal is to reach the end of the course as rapidly as possible, but the skier must pass through a series of gates (indicated by a pair of closely spaced flags). Missed gates count as a penalty against the player's time.

**Tennis.** The player plays the game of tennis. When serving and returning shots, the tennis players automatically swing forehand or backhand as the situation demands, and all shots automatically clear the net and land in bounds.

**Pacman (Ms. Pac-Man).** The player earns points by eating pellets and avoiding ghosts, which contacting them results in losing a life. Eating a "power pellet" causes the ghosts to turn blue, allowing them to be eaten for extra points. Bonus fruits can be eaten for increasing point values, twice per round.

**Breakout** Using a single ball, the player must use the paddle to knock down as many bricks as possible. If the player's paddle misses the ball's rebound, the player will lose a life.

## A.4 Outputs of QA Extraction Module

| Game | Wikipedia | Official Manual |
|------|-----------|-----------------|
| Skiing | **What is the objective of the game?** | |
| | To reach the bottom of the ski course as rapidly as possible. | To reach the bottom of the hill in the fastes time Have fun and god bless!". |
| | **How to succeed in the game?** | |
| | N/A | Learning to control the tips of your skis and anticipating and avoiding trouble. |
| | **How to score at the game?** | |
| | N/A | Elapsed time at the end of the run. |
| | **Who are your enemies?** | |
| | N/A | N/A |
| Pacman | **What is the objective of the game?** | |
| | The player earns points by eating pellets and avoiding ghosts. | To score as many points as you can practice clearing the maze of dots before trying to gobble up the ghosts. |
| | **How to succeed in the game?** | |
| | The player earns points by eating pellets and avoiding ghosts. | Score as many points as you can. |
| | **How to score at the game?** | |
| | The player earns points by eating pellets and avoiding ghosts. | N/A |
| | **Who are your enemies?** | |
| | N/A | Ghosts. stay close to an energy pill before eating it, and tease the ghosts. |
| Tennis | **What is the objective of the game?** | |
| | The first player to win one six-game set is declared the winner of the match. | N/A |
| | **How to succeed in the game?** | |
| | The first player to win one six-game set is declared the winner of the match. | precisely aim your shots and hit them out of reach of your opponent. |
| | **How to score at the game?** | |
| | N/A | The first player to win at least 6 games and be ahead by two games wins the set. |
| | **Who are your enemies?** | |
| | N/A | N/A |
| Breakout | **What is the objective of the game?** | |
| | The player must knock down as many bricks as possible. | Destroy the two walls using five balls To destroy the wall in as little time as possible The first player or team to completely destroy both walls or score the most points To smash their way through the wall and score points |
| | **How to succeed in the game?** | |
| | the player must knock down as many bricks as possible by using the walls and/or the paddle below to hit the ball against the bricks and eliminate them | The first team to destroy a wall or score the most points after playing five balls wins the game To destroy the walls in as little time as possible |
| | **How to score at the game?** | |
| | N/A | Scores are determined by the bricks hit during a game A player scores points by hitting one of the wall's bricks Smash their way through the wall and score points. |
| | **Who are your enemies?** | |
| | N/A | N/A |

Table 5: Table showing the outputs of the QA Extraction module on Wikipedia instructions vs the official Atari manual. The Wikipedia manual is significantly shorter, and contains less information, causing the extractive QA model to repeat answers. However, the overall information, across the 4 questions, captured by the Extraction module is in good agreement across Wiki and Official manuals.

