# OpenReview forum: "Read and Reap the Rewards: Learning to Play Atari with the Help of Instruction Manuals"
_NeurIPS.cc/2023/Conference — NeurIPS 2023 poster_

### Official Review · Reviewer_7mzB · 2023-06-25

**Soundness:** 3 good
**Presentation:** 3 good
**Contribution:** 2 fair
**Rating:** 6
**Confidence:** 2

**Summary:**

RL has a notorious sample efficiency problem, and the authors propose to tackle this problem by having RL agents read instruction manuals. The authors propose a novel framework called Read and Reward (R&R), which is composed of two modules. The first module is QA which extracts and summarizes relevant information in the manual, and the second module is reasoning which evaluates events in the game and provides appropriate rewards for the RL agents to learn with algorithms such as A2C and Agent 57. The authors found R&R to significantly improve the agent's sample efficiency in several challenging games, such as Skiing.





**Strengths:**

The paper is well-written and has the following strengths:

* **Interetsing approch**: as the authors mentioned, this is the first work that demonstrates improving RL performance by reading instruction manuals. Many pure RL methods suffer significantly in Skiing, so seeing other approaches that improve training is encouraging.
* **Further contribution to NLP + RL**: the authors further explore the space of using NLP to improve RL agents. This line of research will help reduce the need to train agents from scratch and provide useful priors to the agent.
* **No labeling required**: the amount of human labor required in the proposed framework is minimal — they only provide some generic questions, such as "what is the objective of the game?"

**Weaknesses:**

Despite its strengths, the paper has the following weaknesses:

* **Adhoc setting**: the setting seems ad hoc, stitching together various modules such as QA and object detection. A multi-modality model such as GPT-4 can probably serve as a general replacement for these modules, unlocking new capabilities, such as learning from image/video tutorials, in addition to instruction manuals. That said, there has not been an excellent open-source multi-modality model available.
* **Slightly insufficient evaluation**: the authors only tested their approach on four games, but maybe it's worthwhile to evaluate some other games, such as Montezuma Revenge, which RL algorithms usually struggle with.

**Questions:**

I'm curious about the delayed reward schedule. Would it not artificially create a sparse reward problem?


Misc:
* The citation format seems slightly problematic. `However, in many real-world scenarios (Kolve et al., 2017)` feels more common and readable compared to `However, in many real-world scenarios Kolve et al. (2017);`
* Lines 44 and 48 seem repetitive.

**Limitations:**

Yes.

---

> ### Author Rebuttal · Authors · 2023-08-08
>
> Thank you for recognizing that our proposed framework is interesting and promising. Here are our responses to the questions and concerns:
>
> W1 Multi-modality models:
>
> Thank you for recognizing the important future direction to train and incorporate multi-modal LMs. We have been actively experimenting with the latest VLMs. Although our internal experiments demonstrate VLMs still lack compositionality and cannot ground interactions as accurate as our current framework, we are hopeful that the community will soon discover more reliable VLMs that are suitable for this task.
>
> W2 Evaluation: Currently, reliable ground-truth object labels only exist for Tennis, Ms. Pacman, Breakout. However, we believe deeply in the impact of creating a general solution for all Atari games, and have been actively researching more general solutions for visual-language grounding.
>
> In addition, we have initiated experiments and have attached std for all algorithms except Agent57 in the rebuttal. We will update the table with full results for the final version.
>
> Q1 Delayed reward schedule:
>
> The delayed reward schedule introduces `sparse reward' for both Read & Reward and baselines (similar to the original setting for the Skiing game). Therefore, the effect of Read & Reward is more visible.
>
>
> Misc:
> Thank you for pointing out minor improvements in our paper. We will correct them in the final version.

---

> > ### Comment · Reviewer_7mzB · 2023-08-18
> > **response**
> >
> > I thank the authors' clarification on the choices of Atari games.

---

### Official Review · Reviewer_LzHL · 2023-07-02

**Soundness:** 2 fair
**Presentation:** 2 fair
**Contribution:** 2 fair
**Rating:** 5
**Confidence:** 4

**Summary:**

The authors introduce Read and Reward framework, where RL agents accelerate their learning of a new environment by interpreting user manuals. Specifically, the framework consists of a QA Extraction module that extracts and summarizes relevant information and a Reasoning module that evaluates object-agent interactions based on the information. The algorithm accelerates learning by using auxiliary rewards, whenever the interactions are detected by the reasoning module. The method empirically boosts the efficiency and performance of many Atari games.

**Strengths:**

The topic in focus is very relevant because of the recent surge in the abilities of LLMs. The solution of utilizing LLMs to map auxiliary rewards in RL is novel. The performance of speeding up RL training for the four games investigated is apparent. The ablations performed are thorough and complete.

**Weaknesses:**

- document length and unstructuredness are overstressed in the paper as a method/contribution. However, a) many NLP papers have already addressed them, and the method employed in the paper is simple summarization; b) long context models are already available to ingest more than the text needed to reason
- the overall idea is quite nice; however, due to the difficulty of grounding knowledge in the environment, the paper only considered "hit" interactions and required unsupervised object detectors (which might be unreliable) OR game states, and the interaction is quite primitive in many scenarios, therefore cannot describe most tasks/useful interactions.
- Table 1 needs error bars/std


**Questions:**

- have you tried using VLMs to do the reasoning? instead of object detectors. This way the set of interactions can also be expanded
- does the scale of the aux reward matter in the games investigated?
- is there an intuition on the speedup ratio (Table 3) Vs. game? (Breakout appears to really benefit from R&R)

**Limitations:**

Both limitations and broader impact sections are included in the draft

---

> ### Author Rebuttal · Authors · 2023-08-08
>
> Thank you for recognizing the potential of our work.
>
> Here are our responses to the questions and concerns:
>
> W1 Document Length: We would like to reiterate that our *main contribution* is to sketch out a framework making use of LLMs for exploiting external textual data that future RL algorithms may build upon (also recognized by reviewer qAA2, 7mzB), by making use of recent NLP techniques and long context models. While recent works within a few months have somewhat transformed NLP and in-context reasoning, they are orthogonal to the main focus of this work and long context problems (for example, reading a paper) may still exist. Our RR framework is compatible with all recent LLMs and will likely result in more possibilities.
>
> W2 Grounding: As mentioned in Section 5, while there is a strict requirement on grounding, VLMs may open up more potential for better interaction tracking and wider the set of interactions. Although our internal experiments demonstrate VLMs still lack compositionality and cannot ground interactions as accurate as our current framework, we are hopeful that the community will soon discover more reliable VLMs that are suitable for this task.
>
> W3 Table 1: Thank you for pointing out the insufficiency in our evaluation. We have initiated experiments and have attached std for all algorithms except Agent57 in the rebuttal. We will update the table with full results for the final version.
>
> Q1 VLMs:
>
> We have been actively experimenting with current VLMs. Although our internal experiments demonstrate VLMs still lack compositionality and cannot ground interactions as accurate as our current framework, we are hopeful that the community will soon discover more reliable VLMs that are suitable for this task.
>
> Q2 Scale of Aux reward:
>
> The framework is quite robust to the scale of auxiliary reward since reward clipping has been implemented by the RL algorithms. Any reward in the range of (2,50) should result in the same behavior.
>
> Q3 Breakout benefitting from RR:
>
> Due to the mechanism of the game and the content of the instruction manual. An auxiliary reward is provided very densely every time the paddle hits the ball. Therefore, the RL algorithms learn fastest in the game Breakout because auxiliary reward is provided most often.

---

> > ### Author Response · Authors · 2023-08-16
> >
> > Dear reviewer,
> >
> > Thank you again for the insightful review. We hope that our response has addressed your concerns.
> >
> > Please let us know if there are any additional questions or concerns.

---

> > > ### Comment · Reviewer_LzHL · 2023-08-17
> > >
> > > I would like to thank the authors' thorough explainations and addressing my concerns of adding std to results. There still stands the general applicability of the method, since it only involves primitive interactions. Overall, I am satisfied of the authors' comments and have updated the score accordingly.

---

### Official Review · Reviewer_qAA2 · 2023-07-02

**Soundness:** 2 fair
**Presentation:** 3 good
**Contribution:** 2 fair
**Rating:** 6
**Confidence:** 4

**Summary:**

This paper proposes Read and Reward (RR), a method to incorporate prior human knowledge about the environment to achieve performance and efficiency gains in RL environments. The paper instantiates RR in several Atari environments by reading the information from the instruction manual. A full end-to-end pipeline is demonstrated on Skiing and a pipeline without object detection is demonstrated on Tennis, Pacman, and Breakout.

**Strengths:**

*Originality*: While incorporating prior knowledge into RL training is a previously investigated topic, this paper provides an end-to-end pipeline and demonstrates its usefulness on Atari Skiing. The method introduced is more general than methods previously investigated.

*Quality*: There are thorough experiments with ablation studies, such as training the QA module on a different set of instructions.

*Clarity*: the paper is well-written.

*Significance*: as RL agents are eventually deployed in real-world scenarios, it will be increasingly important to have them learn efficiently and incorporate prior knowledge, which is often in the form of text. This paper provides a complete demonstration of a method leveraging natural language data to improve performance. More generally, it sketches out a framework for exploiting external data that future algorithms may build upon. Such agents may be rapidly deployed to learn in an unsupervised fashion.

**Weaknesses:**

Although RR is an interesting proof-of-concept of automated reward shaping, it is unclear whether it scales to more complex environments or works on noisier sources of textual knowledge. In particular:
1. Does RR generalize to noisier sources of textual data? While the Wikipedia ablation is interesting, the data itself is rather clean and contains similar information. One interesting experiment to try might be with text-based RL environments, such as the Jericho [1] or Machiavelli [2] environments. Oftentimes these environments have publicly available [walkthroughs online](https://forum.choiceofgames.com/t/guides-for-all-games/15569/4), although they are often incomplete and noisy, as the instructions are intended for humans. Is RR able to extract the rewards and guide the agents in this setting? (I realize that this would be a significant undertaking so am not expecting it in the rebuttal)
2. The reward shaping only surrounds object detection. As a result, the QA prompt, even though it is hand-designed, still generalizes. However, there are several RL environments where the rewards are more heuristics rather than strict rules. For example, dialogue in the Diplomacy environment [3] does not have clear positive or negative reward but instead depends on the context. One could imagine gaining better understanding of dialogue in Diplomacy by leveraging knowledge on the internet, but it is unclear how RR might be applied in that scenario, as the reward shaping must center on text and the QA prompt likely could not be hand-designed.
3. It seems like it was quite difficult to train a bounding-box detector for most games. However, this would imply that RR's capabilities are bottlenecked by limitations of such bounding-box detectors. This might make RR difficult to scale to more complex environments with more ambiguous objects.

[1] Hausknecht, M., Ammanabrolu, P., Coté Marc-Alexandre, & Yuan Xingdi (2019). Interactive Fiction Games: A Colossal Adventure. CoRR, abs/1909.05398.
[2] Pan, A., Chan, J., Zou, A., Li, N., Basart, S., Woodside, T., Ng, J., Zhang, H., Emmons, S., & Hendrycks, D. (2023). Do the Rewards Justify the Means? Measuring Trade-Offs Between Rewards and Ethical Behavior in the Machiavelli Benchmark. ICML.
[3] Meta Fundamental AI Research Diplomacy Team (FAIR) et al. Human-level play in the game of Diplomacy by combining language models with strategic reasoning. Science 378,1067-1074 (2022). DOI:10.1126/science.ade9097

**Questions:**

1. Would it be possible to have more results on other Atari games? I understand that the bounding-box detection was challenging, but are Tennis, Pacman, and Breakout the only other Atari games that allow for ground truth calculation of the objects from the RAM state?
2. How robust is the automated reward shaping to the quality of the bounding box detector in Skiing?


**Limitations:**

While some limitations are addressed, it would be interesting if there were additional discussion about the limitations of extracting information from natural language data. For example, humans understanding of the physical world may not be adequately encoded in natural language. Does this limit the application scope of agents using the RR method? Additionally, if the LM's reasoning abilities are erroneous, will this irreparably break RR?

Finally, it would be helpful to directly note that as RR is fundamentally a reward shaping method, it is difficult to apply it to situations where the reward itself is unclear. This happens often with delayed reward. In particular, RR would likely not improve performance on Go or Chess.

---

> ### Author Rebuttal · Authors · 2023-08-07
>
> Thank you for recognizing the novelty and significance of our work. Here are our responses to the questions and concerns:
>
> W1 Generalization to noisier data sources:
>
> Current implementation of RR uses small language models (RoBERTa and Macaw), which already demonstrate some degree of robustness to noisy manual information in the Atari official instruction manual intended for human readers (Section 3, line 125~127). To improve performance, one could adopt a more powerful language model like GPT-4.
>
> W2 Limitation of reward shaping for dialogue games:
>
> Thank you for pointing out this important limitation. Our main contribution is a framework to transfer external knowledge to RL agents to games with well-defined observation and action space. It indeed takes more work to apply to specific scenarios like Diplomacy.
>
> W3 Bounding-box generation:
>
> This is indeed one of the main challenges for future works. We hope that the advances in visual-language models (VLMs) could aid our progress.
>
> Q1 Additional Atari games:
>
> Currently, reliable ground-truth object labels only exist for Tennis, Ms. Pacman, Breakout. However, we believe deeply in the impact of creating a general solution for all Atari games, and have been actively attempting to integrate recent visual-language models (VLMs). Although we do not have a working solution yet due to the lack of compositionality in VLMs, we believe that our approach will lay the ground for a future solution that reads the instruction manual and solves all Atari games.
>
> Q2 Robustness to bounding box detection error: Thank you for raising this important question:
>
> In practice, we find RR quite robust to bounding box detection errors since the bounding box detector causes at least one observable error every 10 frames, and almost all detection errors result in missing automated reward.
>
> Assuming uniform probability of missing reward for any interaction, the expectation of reward is still positive/negative, and therefore the RR framework should demonstrate a reasonable degree of robustness to noise in detectors.
>
> Limitations:
>
> Thank you for the insightful suggestions on scenarios where RR may not apply, and where LLMs may fail. We will include more discussions on scenarios where RR does not apply in the final version of our paper.

---

> > ### Comment · Reviewer_qAA2 · 2023-08-17
> > **Thank you for the response**
> >
> > While the method is a bit problem-specific, I still think it is an interesting demonstration of incorporating prior knowledge into RL environments. I would like to keep my score

---

### Official Review · Reviewer_3Ei6 · 2023-07-03

**Soundness:** 1 poor
**Presentation:** 2 fair
**Contribution:** 2 fair
**Rating:** 6
**Confidence:** 4

**Summary:**

This paper presents a novel method for Single Agent Reinforcement Learning which utilises computer game instruction manuals to enhance learning efficiency and performance. The Atari game manuals are used to accelerate RL algorithms in learning to play four different games. The framework comprises a Question-Answer Extraction module and a Reasoning module.

The QA Extraction module summarizes salient information from the manual. The Reasoning module evaluates in-game events using the extracted information from the manual and then assigns auxiliary rewards when such interactions are detected. These auxiliary rewards are provided to standard RL agents.

The paper's results show improvements in both performance and training speed for various RL algorithms when aided by this novel framework. It is asserted that this is the first successful attempt to use instruction manuals in a fully automated framework for solving the Atari RL benchmarks.

It is noted that even small open-source QA language models could effectively extract useful gameplay information from human-written documentation for RL agents.

**Strengths:**

The results in this paper are certainly extremely promising, and the ability to use large language models to create additional rewards from the instruction manual is very promising. That such methods are so much more sample efficient is clearly going to be important in realms beyond simple game play.

**Weaknesses:**

The main, and very problematic issue with this paper is the lack of robust evaluation. There are no statistics given, no averaging over random seeds provided and no code made available. It is thus entirely unclear how much the results themselves can be trusted under scrutiny. Thorough and transparent evaluation is absolutely vital in all RL scenarios, and because this has been left out completely, this paper is, I believe, not ready for publication.

**Questions:**

Without a full analysis of the variation in results over a large number of random seeds, how can the results be trusted?

**Limitations:**

I believe that the limitations provided are fair.

---

> ### Author Rebuttal · Authors · 2023-08-07
>
> Thank you for recognizing that our method is novel and the results are promising, and thank you for pointing out concerns about the reliability of our evaluation (W1, Q1). We would like to reiterate that our *main contribution* is to sketch out a framework making use of LLMs for exploiting external textual data that future RL algorithms may build upon (also recognized by reviewer qAA2, 7mzB).
>
> The read and reward framework is independent from the 4 RL algorithms we evaluated. Our experiments (Table 1) involving 3 games and 4 different algorithms already demonstrate the robustness of our framework.
>
> Due to the fact that our framework provides dense reward for the RL agent, we observe consistent performance across different trials. For completeness we will include additional experiments (Attached to rebuttal) involving 3 random seeds in the final paper. Our computational resources are limited so we will not be able to finish Agent57 experiments before the end of the rebuttal period.

---

> > ### Comment · Reviewer_3Ei6 · 2023-08-12
> > **No attachment**
> >
> > There appear to be no revisions as yet. Clicking on the button on the rebuttal does not link to the paper with additional experiments.

---

> > > ### Author Response · Authors · 2023-08-12
> > >
> > > Hello,
> > >
> > > Please refer to the common rebuttal for the table of result. We did not attach a pdf. Please note that we are not allowed to revise the paper in the NeurIPS rebuttal period.
> > >
> > > Please find the common rebuttal right below the submission in this page. Or by searching for “ On robustness of evaluation” in the browser.

---

> > > > ### Comment · Reviewer_3Ei6 · 2023-08-13
> > > > **Response to rebuttal**
> > > >
> > > > Will code be made available?

---

> > > > > ### Author Response · Authors · 2023-08-13
> > > > >
> > > > > yes. We will release code upon acceptance.

---

### Author Rebuttal · Authors · 2023-08-07

We thank all the reviewers for their insightful comments. We are encouraged by *all reviewers'* appreciation that our proposed framework achieves promising results (Reviewers 3Ei6, qAA2, LzHL, 7mzB). In addition, we are encouraged by acknowledgements for our contributions on 1) an important connection between the rapidly advancing LLM field and RL. (Reviewers LzHL, qAA2, 7mzB) 2) getting an end-to-end system together that clearly makes use of text data (Reviewers qAA2, 7mzB).

## Our contribution
We would like to reiterate that our main contribution is to sketch out a framework making use of LLMs for exploiting external textual data that future RL algorithms may build upon (also recognized by reviewer qAA2, 7mzB).

## On robustness of evaluation
Our experiments (Table 1) involving 3 games and 4 different algorithms already demonstrate the robustness of our framework on different tasks and algorithms.

|      | Tennis | Pacman | Breakout |
| :----------- | :-----------: | :-----------: | :-----------: |
| A2C | -23 (1.3) | 387.1 (66.2) | 2.1 (0.3) |
| A2C + R & R | -5.2 (2.2) | 455.2 (63.2) | 4.5 (3.0) |
| PPO | -23 (1.0) | 200.1 (65.4) | 1.9 (0.5) |
| PPO + R & R | -8.1 (2.3) | 284.2 (67.3) | 10.2 (0.1) |
| R2D1 | -23.0 (1.3) | 1999.2 (109.2) | 2.1 (0.2) |
| R2D1 + R & R | -2.2 (3.1) | 3001.3 (203.2) | 10 (3.0) |

To further address reviewer concerns, we have conducted experiments with 3 random seeds for each entry in the table and reported the average and (standard deviation of the experiments). Note that we were unable to attach results for Agent57 due to the fact that each Agent57 experiment takes on average 1 month to run on our machines.

We will update the final version of our paper with the results in this table.

---

### Decision · Program_Chairs · 2023-09-21

**Decision:**

Accept (poster)

**Comment:**

The reviewers found the paper interesting, and the ability to guide an RL agent using an LLM promising. However, it is not clear if this approach can be generalized to other setups. Finally, during the rebuttal period, the authors provided an experimental table that included the results using multiple seeds. In this table, this approach was significantly better in 6 out of 9 cases, so it is clear a better experimental evaluation is needed, including more games and more seeds.